# The first demonstration of entirely roll-to-roll fabricated perovskite solar cell modules under ambient room conditions

Hasitha C. Weerasinghe[1,8], Nasiruddin Macadam[2,8], Jueng-Eun Kim [1,3,8], Luke J. Sutherland[1,3], Dechan Angmo[1], Leonard W. T. Ng[1,2,4], Andrew D. Scully [1], Fiona Glenn[1], Regine Chantler [1], Nathan L. Chang [5], Mohammad Dehghanimadvar [5], Lei Shi[5,6], Anita W. Y. Ho-Baillie [5,7], Renate Egan [5], Anthony S. R. Chesman [1], Mei Gao [1], Jacek J. Jasieniak [3] ✉, Tawfique Hasan [2] ✉ & Doojin Vak [1] ✉

The rapid development of organic-inorganic hybrid perovskite solar cells has resulted in laboratory-scale devices having power conversion efficiencies that are competitive with commercialised technologies. However, hybrid perovskite solar cells are yet to make an impact beyond the research community, with translation to large-area devices fabricated by industry-relevant manufacturing methods remaining a critical challenge. Here we report the first demonstration of hybrid perovskite solar cell modules, comprising serially-interconnected cells, produced entirely using industrial roll-to-roll printing tools under ambient room conditions. As part of this development, costly vacuum-deposited metal electrodes are replaced with printed carbon electrodes. A high-throughput experiment involving the analysis of batches of 1600 cells produced using 20 parameter combinations enabled rapid optimisation over a large parameter space. The optimised roll-to-roll fabricated hybrid perovskite solar cells show power conversion efficiencies of up to 15.5% for individual small-area cells and 11.0% for serially-interconnected cells in large-area modules. Based on the devices produced in this work, a cost of ~0.7 USD W$^{-1}$ is predicted for a production rate of 1,000,000 m$^2$ per year in Australia, with potential for further significant cost reductions.

Organic-inorganic hybrid perovskite solar cells (PeSCs) are a promising next-generation photovoltaic (PV) technology that has a demonstrated power conversion efficiency (PCE) of 26.1%[1]. Despite the record efficiencies being competitive with the market incumbent technology, crystalline Si PV with 26.8% PCE[1], numerous challenges must be addressed for PeSCs to be realised in real-world applications. Foremost is the need to translate small-area lab-scale cells, which are often fabricated using materials or methods that are not economically viable or

---

[1]Flexible Electronics Laboratory, CSIRO Manufacturing, Clayton, VIC 3168, Australia. [2]Cambridge Graphene Centre, University of Cambridge, Cambridge CB3 0FA, UK. [3]Department of Materials Science and Engineering, Monash University, Clayton, VIC 3800, Australia. [4]School of Materials Science and Engineering (MSE), Nanyang Technological University (NTU), 50 Nanyang Ave, Block N4.1, Singapore 639798, Singapore. [5]School of Photovoltaic and Renewable Energy Engineering, University of New South Wales, Sydney, NSW 2052, Australia. [6]Foshan Xianhu Laboratory of the Advanced Energy Science and Technology Guangdong Laboratory, Foshan, China. [7]Sydney Nano and School of Physics, Faculty of Science, The University of Sydney, Sydney, NSW 2006, Australia. [8]These authors contributed equally: Hasitha C. Weerasinghe, Nasiruddin Macadam, Jueng-Eun Kim. ✉e-mail: jacek.jasieniak@monash.edu; th270@cam.ac.uk; doojin.vak@csiro.au

scalable, to large-area devices produced by high-volume, low-cost manufacturing methods. As shown by other solar PV technologies with high PCEs, such as inorganic multi-junction or GaAs cells, a failure to lower production costs will prevent PeSCs from making an impact in the marketplace[2,3].

A key difference between PeSCs and conventional inorganic PV technologies is the potential for low-cost and low-energy manufacturing using solution-based industrial processes, such as spray[4–6], blade[7–9] and slot-die (SD) coating[10–15]. Recent advances in large-area glass-based PeSCs have resulted in promising efficiencies of up to 25.8%[16–19]. However, these devices have been produced using discrete sheet-to-sheet processing, utilise vacuum-based evaporation steps, and employ subtractive laser-patterning to achieve interconnections for large-area modules. These requirements will add challenges in lowering the cost of large-scale production. In contrast, flexible PeSCs enable high-volume and high-throughput manufacturing using continuous roll-to-roll (R2R) manufacturing techniques[20–22]. The lightweight and physical flexibility of flexible PeSCs also offer the prospect of solar PV panels having high specific power (power-to-weight ratio), which is highly desirable for emerging applications, including space[3], vehicle-integrated PV, and building-integrated PV[2,23]. However, the process of manufacturing PeSCs on a continuously-moving flexible plastic substrate imposes several technical challenges, particularly time and temperature processing limitations[24].

Beyond advancing the manufacturing process, replacing the high-cost components in the solar cell architecture with cheaper alternatives while retaining comparable performance remains a persistent challenge. The highest cost component is the vacuum-processed Au electrode, followed by commercially produced transparent conductive electrodes (TCEs). Vacuum deposition is costly, and the nature of the process is unsuitable for use with a conventional R2R manufacturing line. There have been several reports of solution-processed back electrodes in glass-based devices[25–27], but their processing involved a prolonged high-temperature step that is neither compatible with flexible plastic substrates nor suitable for R2R-based upscaling due to time constraints in the continuous process. Due to these technical challenges, the first example of a small-area PeSC (0.09 cm² active area) having all layers deposited on a flexible plastic substrate using R2R processes was only very recently reported (in February 2023)[28] with individual cells displaying PCEs of up to 10.8%. While the first report marked a significant milestone in the field, the efficiency was still far from that of typical research cells and only small cells were demonstrated.

Here we report the fabrication of entirely R2R-printed individual PeSCs with a record-high 15.5% PCE. We also report the first demonstration of PeSC modules produced using only industry-relevant R2R fabrication techniques, and under ambient room conditions. This was achieved by developing: (i) a robust and scalable deposition technique, (ii) perovskite-friendly carbon inks to replace vacuum-based electrodes, and (iii) a R2R-based high-throughput experimental platform as illustrated in Fig. 1a. The latter mimics manufacturing processes to produce and test thousands of research cells a day. This allowed the seamless translation from the miniature factory to the full-scale R2R fabrication of PeSC modules (~50 cm² active area) exhibiting up to 11% PCE. The future prospects of the printed PeSCs are evaluated by considering manufacturing costs for various production scenarios calculated using cost models based on the production methods and materials used in the present work, together with the resulting device efficiencies.

## Results

### Control of perovskite crystallisation for upscaling

Although spin coating has been widely adopted to produce efficient PeSCs, the deposition and drying parameters are significantly different in R2R production. As such, it is necessary to initially develop processing conditions using R2R or R2R-compatible methods. The introduction of the printing-friendly sequential deposition (PFSD) technique by select co-authors of this work in 2017[13] enabled the demonstration of the first PeSCs comprising R2R-deposited electron-transport layer (ETL), light-absorbing layer, and hole-transport layer (HTL), with up to 11% PCE achieved for a small-area device. Since then we also developed more facile single-step deposition techniques via the introduction of various additives such as polymers, ammonium salts, and 2D organic cations together with heating and nitrogen blowing[12,29–33], and investigated R2R techniques reported by others[10,11,15,24,34]. Although it was possible to produce the perovskite layer in a single-step deposition, we found no approach that significantly outperforms PFSD for R2R-based upscaling.

The PFSD approach is described as 'printing friendly' due to its robustness and reliability under ambient conditions, and the absence of time-consuming processing steps. The key to PFSD is adding organic cations at a loading of less than 50 mol% of PbI$_2$, far below the stoichiometric amount required to form perovskite crystals. This strategy retards crystallization and the precursor thin-film behaves like an amorphous material with much better film-forming properties than crystalline analogues. When additional organic cation is subsequently deposited, the reactive amorphous-phase film quickly converts to a perovskite without needing to remove the additive as it becomes a part of the perovskite. This allows the conversion to be completed on a time scale suitable for R2R processing.

Our further development of the PFSD method resulted in up to 17.9% PCE from R2R-fabricated PeSCs with vacuum-deposited Au electrodes, as discussed in Supplementary Note 2. One significant innovation in the PFSD technique is the introduction of a shallow-angle blowing technique, (Fig. 2a) as opposed to the conventional blowing technique applied at a right angle, used to fabricate high-quality perovskite films involving blowing gas across the substrate[35]. While effective at a laboratory scale, creating a highly uniform gas flow over a large area is challenging. The shallow-angle blowing on the edge of a roller is a simple but effective way to upscale this process, and the angle of incidence can be easily adjusted to levels that can approach zero degrees by simply changing the blower-head position. Utilising this approach, the SD-coated wet films are not deformed by an aggressive air flow before entering the well-defined solidification zone. This significantly reduces the amount of crystalline defects, and amorphous glassy films can be fabricated that are then converted instantly to a perovskite when the methyl ammonium iodide (MAI) solution is deposited, as shown in Supplementary Fig. 1.

Figure 2b shows the improved quality of the perovskite film fabricated with the edge-blowing technique. The inset photograph shows the flawless mirror-like perovskite film continuously fabricated under ambient conditions (40–50% relative humidity). X-ray diffraction (XRD) analysis of the film does not indicate the presence of PbI$_2$ crystals, which would be evidence of ion migration followed by inhomogeneous local concentration in the solidification process. Shallow-angle blowing produced an intermediate layer that appeared to be amorphous or comprised of small enough grains that allowed for rapid and complete conversion to perovskite upon MAI deposition. Notably, this could not be examined directly as the reactive and unstable intermediate films continuously change upon exposure to air. Scanning electron microscope (SEM) images in Supplementary Fig. 2 show more homogenous films with compact grains of the shallow-angle-blown sample compared to the right-angle-blown sample. The introduction of the shallow-angle blowing not only improved the quality of the perovskite and the reliability of device performance, but also enhanced humidity tolerance (as discussed later), making the PFSD approach a suitable technique for low-cost manufacturing.

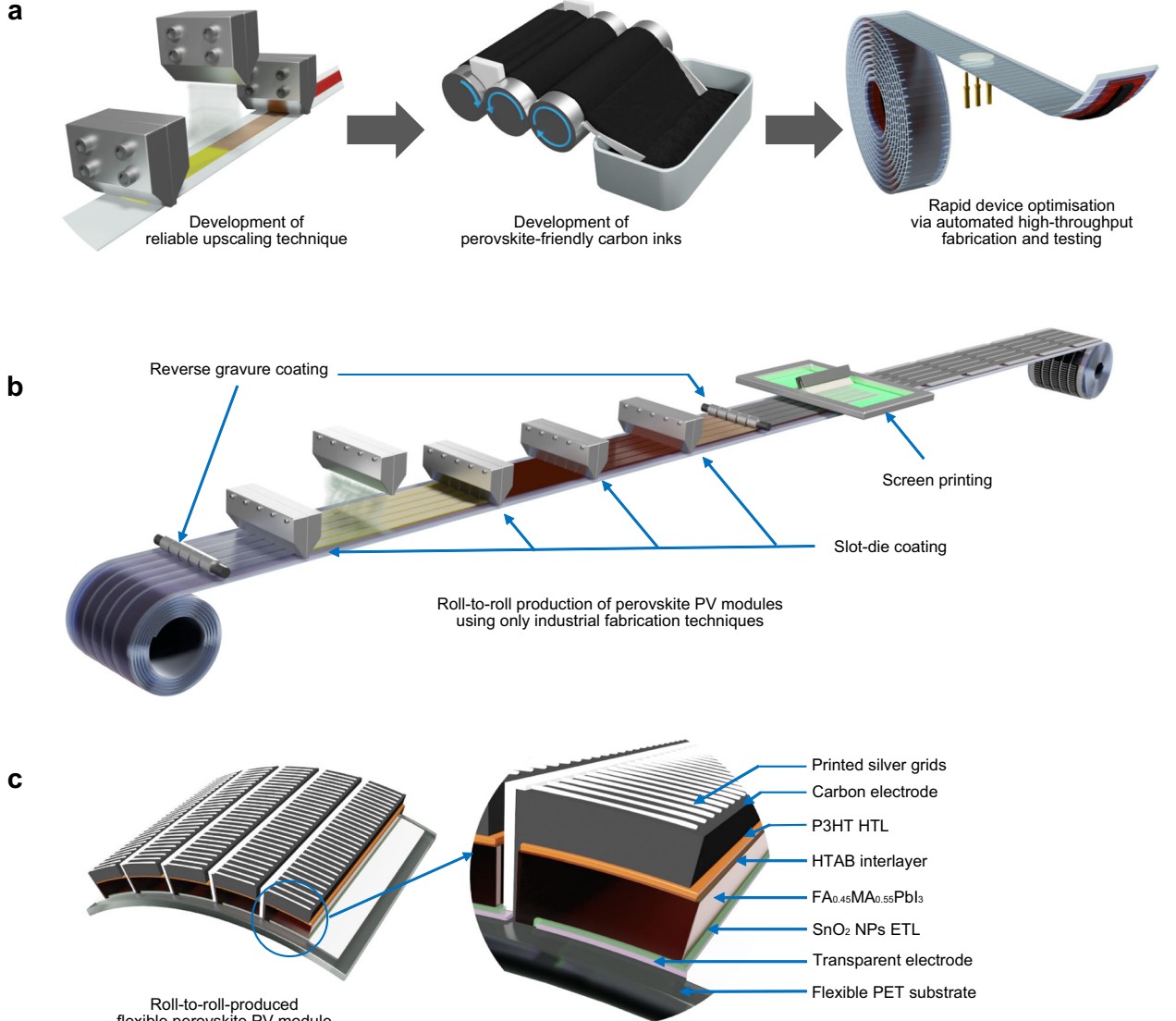

**Fig. 1 | Schematic illustration of the workflow of this work. a** A reliable SD coating process and a perovskite-friendly carbon ink are developed to enable vacuum-free perovskite PV production. The carbon ink is upscaled using a three-roll mill and used to optimise device parameters by fabricating and testing numerous research cells using an automated roll-to-roll research platform. **b** Schematic illustration of roll-to-roll production of modules using SD coating, reverse gravure (RG) coating and screen printing. **c** The detailed structure of the series connected module, which is fully roll-to-roll fabricated on commercially available transparent electrodes.

## Automated, ambient, and vacuum-free device fabrication

The discovery of R2R-printable electrodes for PeSCs has long been a critical challenge in the realisation of fully R2R-fabricated vacuum-free cells. To date, most R2R-fabricated PeSCs in the literature incorporate vacuum or batch-processed back electrodes. The first fully R2R-fabricated PeSC was reported recently using a printed carbon electrode, achieving a PCE of 10.8%[28]. The efficiency was significantly lower than vacuum-based counterparts, suggesting performance degradation caused by the carbon ink. Therefore, we developed perovskite-friendly carbon inks and trialled them alongside commercially available carbon pastes, as discussed in Supplementary Note 3.

Replacing this vacuum process, which is not only costly but also time consuming, has the additional benefit of creating a new avenue for experimental optimisation. While such an approach improves throughput for conventional R2R systems with manual operation[36], its full potential is realised with the development of a programmable R2R SD coater for unmanned operation, allowing for the fabrication of thousands of unique PeSCs daily. Manual characterisation of this many

cells is not practical. We therefore developed an automated R2R tester to test over ten thousand solar cells a day. Device parameters were automatically calculated and saved online, permitting the analysis of thousands of solar cells in minutes, rather than hours or days. The custom-built R2R research tools are shown in Fig. 3a, b, and demonstrations of the system can be seen in the Supplementary Information. Figure 3c shows the device layout and testing setup for the high-throughput testing of R2R-fabricated solar cells.

This high-throughput experimental platform enabled us to explore the extensive fabrication parameters of vacuum-free PeSCs to rapidly identify the optimal conditions. Figure 3d–h show an example of the high-throughput R2R experiment, in this case being used to optimise deposition parameters for PbI$_2$ with 45 mol% formamidinium iodide (FA) and MAI solutions and to identify composition-dependant device parameters. Three PbI$_2$ conditions were selected to fabricate perovskite layers of about 600 nm to 1000 nm thickness. This range is somewhat thicker than typical vacuum-deposited electrode devices due to the absence of a mirror effect from the carbon-based back

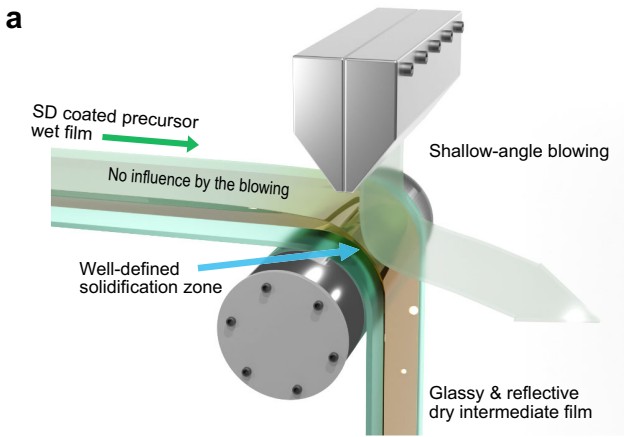

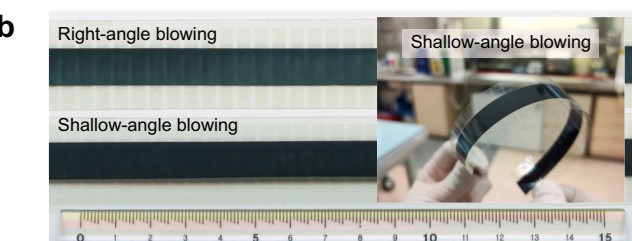

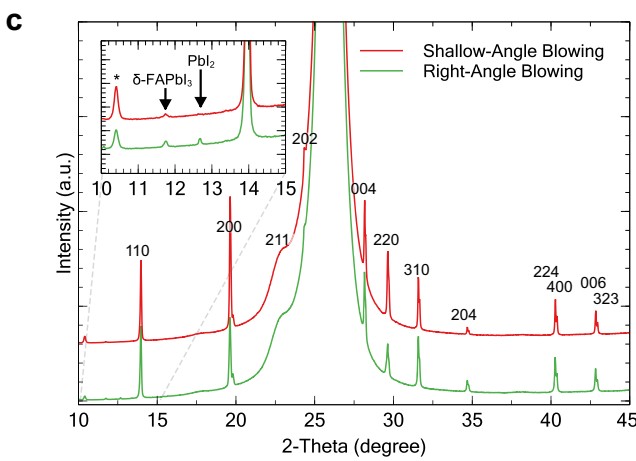

**Fig. 2 | Reliable fabrication of high-quality perovskite film by the edge-blowing technique. a** Schematic illustration of the edge-blowing technique in the roll-to-roll process. **b** An image (reflection mode) of perovskite films after MAI deposition. The upper sample shows a face-blowing sample with a hazy surface and the lower sample shows an edge-blowing sample with a darker and more uniform surface. Inset shows the mirror-like perovskite film fabricated with the shallow-angle blowing. **c** XRD data of the perovskite films produced using the two blowing methods. (The large peak at 27° is PET and the peak at 10.4° is hydrated perovskite of the air-exposed sample.) Source data are provided with this paper.

electrode. Also, a perfectly matched stoichiometry is not necessarily the best formulation in the $FA_{0.45}MA_{0.55}PbI_3$ system, as it can benefit from either a slight excess of lead[37–39] or a cation-excessive composition[40]. Therefore, the ability of SD coating to give quantitative control over the amount of material deposited allowed for the amount of MAI present in the perovskite layer to be varied from slightly cation deficient (lead-excessive composition) through to stoichiometric and slightly excessive compositions for each $PbI_2$ condition. The MAI-deposition flow rate was varied between 30 and 100 μL min$^{-1}$ in 10 μL min$^{-1}$ intervals. Figure 3h shows the deposition parameters together with the PCEs of 1600 consecutively fabricated PeSCs with 20 deposition parameters depending on the position along a 9 m-long

substrate. The PeSCs were obtained in a roll form and the roll was transferred to the automatic R2R PV tester shown in Fig. 3b.

Figure 3d–h show statistical device parameters obtained from 80 cells for each condition. A thickness-tolerance hole transport material, poly[(2,5-bis(2-hexyldecyloxy)phenylene)-alt-(5,6-difluoro-4,7-di(thiophen-2-yl)benzo[c]-[1,2,5]thiadiazole)] (PPDT2FBT)[41] (further discussion on the material choice can be seen in Supplementary Note 3), which was used to screen carbon inks was used as an HTL in these devices. The devices with an MAI content close to the stoichiometric amount show better performance than others. The thinnest condition (16 μL min$^{-1}$) shows the best performance at the stoichiometric amount and performance decreases rapidly with an excess of MAI or $PbI_2$. Thicker films show more interesting behaviour; MAI-deficient films show better fill factor (*FF*) with narrow performance variations, while films with excess MAI show higher short-circuit current ($J_{sc}$). Most importantly, these results demonstrate the utility of the high-throughput experimental platform by identifying a composition-dependent performance trend in just one day.

A significant improvement in fully R2R-fabricated cells was achieved by introducing a new hole-transport layer (HTL) system. Poly(3-hexylthiophene) (P3HT) is a simple, widely used conjugated polymer with good light-harvesting and charge-transporting properties[42] that lends itself to low-cost mass production. While the polymer alone shows relatively poor performance as an HTL, it shows promising performance when combined with *n*-hexyl trimethyl ammonium bromide (HTAB)[43], which passivates the surface traps of the perovskite layer and also provides anchoring points for the hexyl side chain of P3HT to self-assemble in the preferred molecular orientation. Despite such advantages, this HTL system has never been used with R2R-fabricated PeSCs, likely due to the technical difficulty of forming an ultra-thin HTAB skin on perovskite layers utilising a scalable deposition technique. The fabrication of the HTAB skin is achieved through delicate control of surface reactivity by adjusting the ratio of relatively nonpolar and less reactive chlorobenzene to 2-propanol solvents. A uniform P3HT layer was achieved by heating the substrate to 45 °C, which lowers the surface tension of the polymer solution and promotes the self-assembly of P3HT on the HTAB surface. Without substrate heating, P3HT formed poor films on HTAB intermittently, as shown in Fig. 4a.

The HTAB-P3HT HTL clearly outperformed PPDT2FBT, as shown in Fig. 4b. The devices not only showed higher performance but also improved reliability, indicated by the narrower distribution in the histogram. All device parameters, including the hysteresis factor of the HTAB-P3HT-based devices, can be seen in Fig. 4c–g. The figures show the parameters of the unfiltered 240 cells that were consecutively fabricated using the automated R2R SD coater. The substrate consists of blocks of 40 electrodes, separated by a 2 cm gap which can be seen in the data. The experiment was carried out under uncontrolled ambient conditions on a day with high relative humidity (~60% RH), demonstrating the robustness of the manufacturing process. In general, while the best devices were obtained on days with low humidity (30-40% RH), reliable production of PeSCs with an average PCE of ~13% was confirmed, regardless of the humidity in the lab. The J-V curve and IPCE spectrum of the best-performing device are shown in Fig. 4h, i, respectively, with 15.5% PCE, 19.9 mA cm$^{-2}$ $J_{sc}$, 76.1% *FF* and 1.02 V $V_{oc}$. The IPCE spectrum shows good agreement with a calculated current density of 19.4 mA cm$^{-2}$.

## Transitioning from cells to modules
The optimised device fabrication parameters were used to produce large-area modules. Since all processes were developed by scalable deposition methods, modules could be fabricated in the same way using larger SD heads, as shown in Fig. 5a, and a 10 cm wide substrate with a pre-patterned TCE. The SD heads have five channels (a detailed structure can be seen in Supplementary Fig. 3), and hence the

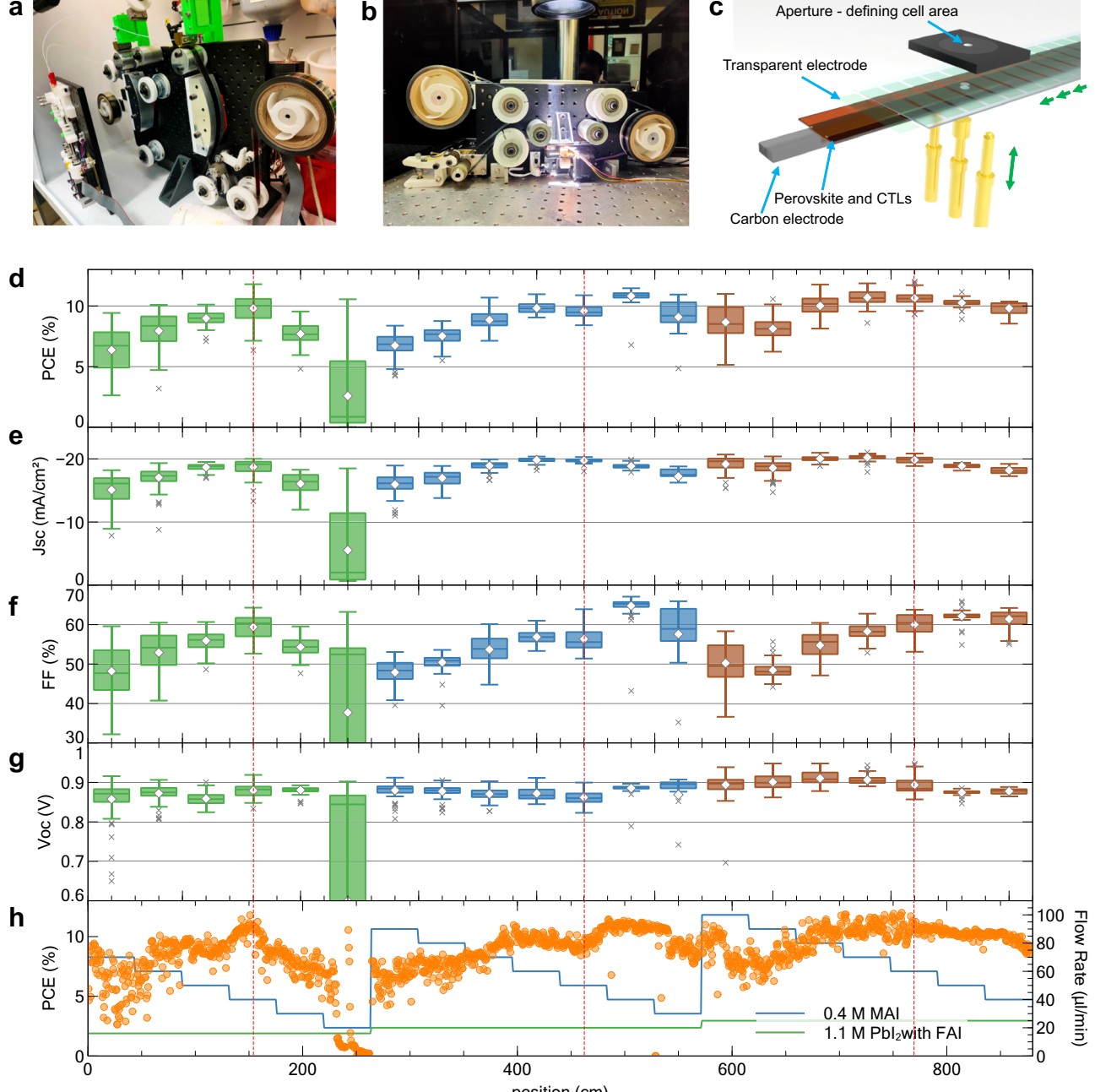

**Fig. 3 | Automated fabrication of 1600 PeSCs in one experiment. a** The custom-built R2R SD coater used for the fabrication of thousands of unique solar cells in a day. **b** The custom-built automatic solar tester with a daily throughput of over 10,000 cells. **c** Schematic illustration of the automatic testing setup. **d**–**h** An example of a high-throughput R2R experiment to screen deposition parameters of the perovskite layer of PeSCs with a configuration: flexible TCE/SnO$_2$/FA$_{0.45}$MA$_{0.55}$PbI$_3$/PPDT2FBT/carbon electrode. Statistical (**d**) PCE, (**e**) $J_{sc}$, (**f**) $FF$, and (**g**) open-circuit voltage ($V_{oc}$) of 80 devices (along 44 cm substrate for each condition) for 20 different deposition parameter combinations of PbI$_2$ with 45 mol % FAI and MAI solutions. (centre line, median; diamond, mean; box limits, upper and lower quartiles; whiskers, 1.5× interquartile range; linecross, outliers) (**h**) PCE of 1600 consecutively fabricated PeSCs and deposition parameters along the position of the flexible film. Red dashed lines indicate the MAI amount that is nearest to the theoretical stoichiometric amount. The error bars in the box charts represent 1.5 times the interquartile range. Source data are provided with this paper.

optimised flow rate was simply multiplied by five to produce five-cell modules. Supplementary Fig. 4 shows images of high-quality perovskite stripes produced continuously using the R2R SD coating method. The HTAB and P3HT layers were also deposited at a flow rate five times higher than the optimised flow rate for single-stripe coating.

For the R2R-deposited electrode, the carbon ink was deposited using the reverse gravure (RG) technique[44]. The modules were completed by R2R screen printing a silver paste on the carbon film using an industrial R2R screen printer, shown in Fig. 5c and Supplementary

Fig. 5. The printed silver was used to form charge-collection grids and interconnect the 5 individual cells, as illustrated in the inset of Fig. 5d. It was crucial to incorporate additional conductive elements alongside the carbon layer, which had a sheet resistance of approximately 800Ω sq$^{-1}$. Supplementary Fig. 11 illustrates that cells without a grid design exhibited significantly poorer performance compared to those with grids. Nonetheless, excessively covering the screen-printed silver led to a decline in performance, most likely due to solvent damage to the underlying layers. Consequently, we designed the silver pattern to

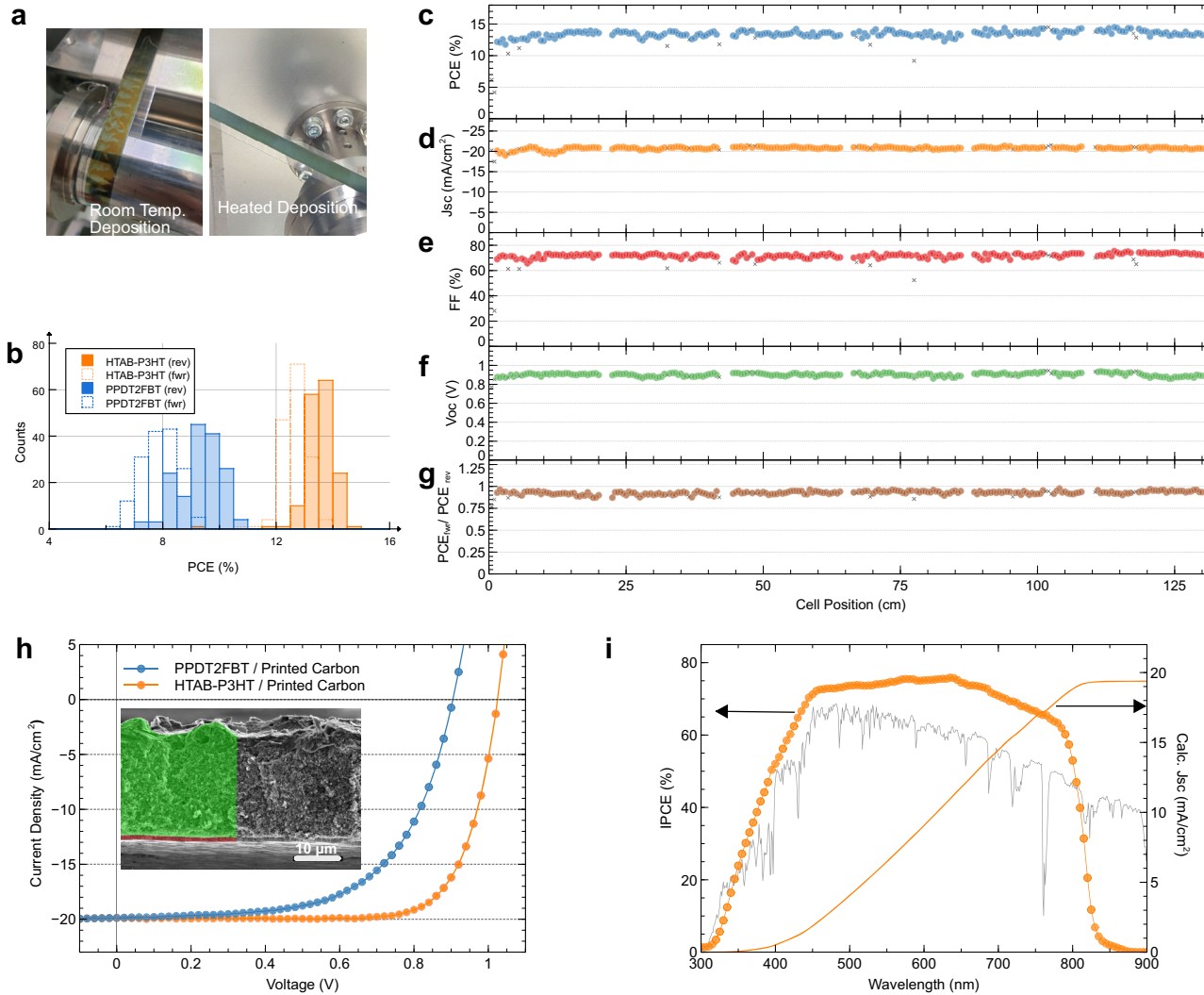

**Fig. 4 | Reliable fabrication of fully R2R-processed PeSCs. a** R2R SD-coated P3HT films on perovskite films with and without gentle heating of the coating stage. **b** Histograms of fully R2R-fabricated PeSCs with a configuration of flexible TCE/$SnO_2$/$FAI_{0.45}MA_{0.55}PbI_3$/HTAB-P3HT or PPDT2FBT/carbon electrode. **c** PCE, (**d**) $J_{sc}$, (**e**) *FF*, (**f**) $V_{oc}$, and (**g**) hysteresis factor (forward-scan PCE divided by reverse-scan PCE) of the 240 consecutively fabricated HTAB-P3HT-based devices. The devices were fabricated in uncontrolled ambient conditions on a high-humidity (~60% RH)

day. (**h**) Current density-voltage (J-V) curves of vacuum-free PeSCs comparing two HTLs. Inset shows an SEM cross-section image of the device with printed carbon (marked in green). **i** Incident-photon-to-current efficiency (IPCE) spectrum and calculated current density of the device with HTAB-P3HT HTL and printed carbon electrode. The AM 1.5 G spectrum used to calculate the current is shown in grey. Source data are provided with this paper.

achieve minimal coverage while maintaining adequate conductivity, at least surpassing that of the front electrode, for efficient charge collection. We determined that a 0.2 mm line with a 180 mesh screen provided the finest pattern that we could consistently print onto the carbon surface, and this parameter was adopted for the module.

The interconnection of cells was formed through gaps between the stripe patterns at the same time as the grid. The active area is 49.5 cm² (1.1 cm × 9 cm × 5 cells) and the geometric fill factor (GFF), as defined as the cell area over total area (cell area + gap area)[45] of the modules, is 75%. The GFF is somewhat lower than that of laser scribed modules with a demonstrated GFF of up to 99%[46] due to the inherent limitation of the stripe-pattern approach. However, it's worth noting that the laser scribing process may not be suitable for the high-throughput, cost-effective manufacturing of PeSCs. Therefore, our next challenge lies in developing modules with higher GFFs and larger areas while continuing to enhance cell efficiencies through scalable processes.

The entirely R2R-fabricated modules demonstrated up to 11.0% active-area-based PCE with 192 mA current output, 62.3% *FF* and 4.59 V

$V_{oc}$ in a reverse scan and 9.96% PCE in a forward scan. This efficiency is lower than the small cells, likely due to the loss of *FF* caused by the high resistance of the TCE and partial solvent damage incurred during the screen-printing process. Nonetheless, this report marks a significant milestone in the development of fully R2R-fabricated PeSCs.

The efficiencies obtained in this work are compared with the previous records of various PeSCs, as summarised in Fig. 5e and detailed in Supplementary Table 1. Flexible PeSCs have always shown inferior performance compared to their glass-based counterparts due to the intrinsic performance limitation of the flexible TCEs. Therefore, batch-processed flexible PeSCs are a more suitable benchmark for R2R PeSCs as they share the same intrinsic properties. While the performances of R2R devices still trail those of batch-processed analogues, this work demonstrates significant progress towards achieving high-efficiency flexible devices. Considering the low-cost nature and the scalability of the R2R-printed carbon electrode, achieving over 15% PCE represents a major milestone in the development of this technology. Through market surveys and considering advantages in the form factor, we established that R2R PeSCs could become competitive in the

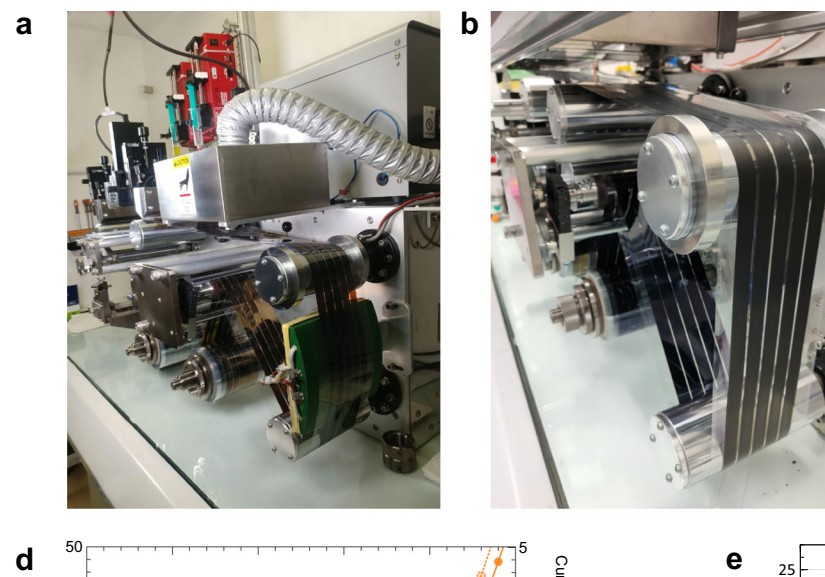
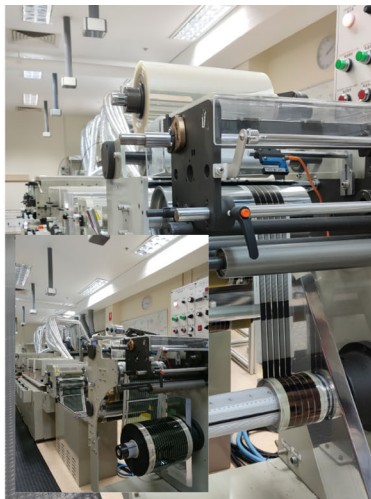

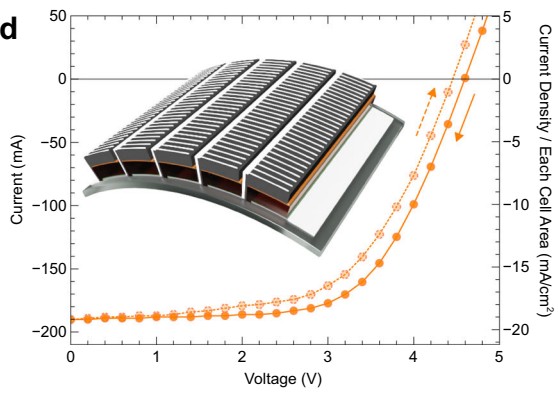
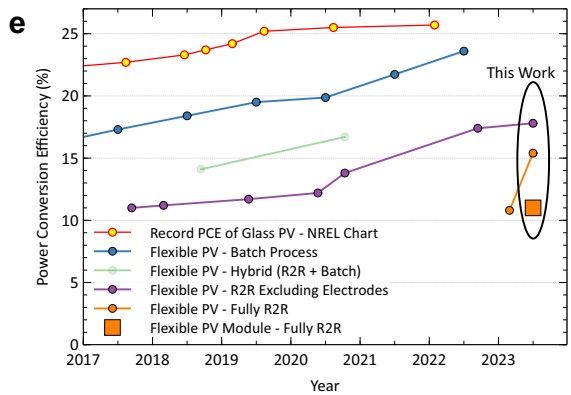

**Fig. 5 | R2R production of perovskite PV modules. a** Image showing the deposition of the perovskite layer using a 5-channel SD coater. The deposition of non-stoichiometric perovskite precursor solution, blow-drying, and the conversion to perovskite by SD coating the MAI solution were carried out in one coating run. **b** Image showing the RG-coated carbon film on the TCE/ETL/perovskite/HTL stack. **c** Image of the R2R-screen-printed perovskite PV modules. An industrial off-the-shelf screen printer was used for this production. Inset shows the capacity of the machine. The photo was taken when the printer was used to produce 30 cm × 500 m organic PV modules. **d** Current-voltage curves of a R2R produced module. Inset shows a schematic illustration of the module structure. **e** Record efficiencies of various perovskite solar cells. Details can be seen in Supplementary Table 1. Source data are provided with this paper.

portable PV market at >10% PCE[47]. Therefore, the demonstration of an 11% R2R-fabricated module is a significant step forward in commercialising this technology. However, the printed silver used in this work may not be suitable for long-term operation for commercial applications due to the corrosion issue. The next challenge would be developing a perovskite-friendly conductive carbon ink that is at least as conductive as TCEs to produce efficient silver-free PeSC modules.

### Towards ultra-low-cost manufacturing

This work aimed to develop low-cost manufacturing technologies for PeSCs. Therefore, we developed a cost model based on our previous work[47] which implemented the new materials, processes and device configurations used in this work, as shown in Fig. 6a. Further details of the cost model can be seen in Supplementary Note 4. In addition to the demonstrated devices, a model device architecture (Sequence C) is also considered to predict the potential for further cost reduction by eliminating the remaining high-cost components, i.e., commercial TCEs and silver grids.

Figure 6b, c show the cost fraction of each functional layer and corresponding capital costs for sequences A and B. For the vacuum-deposited electrode (Seq. A), a combination of the gold material and the equipment purchase and running costs of a R2R evaporator, is the highest cost component, followed by the commercial TCE. Other significant material costs include the encapsulation materials and the HTL

whilst the costs of the ETL and perovskite are negligible in comparison. The fully printed configuration, Seq. B, shown in Fig. 6c, shows a significant reduction in the back electrode cost, resulting in only two high-cost components; the commercial TCE, and the encapsulation material.

Figure 6d, e show the production cost of encapsulated flexible perovskite solar modules per unit area (m²) and peak power (W_p), respectively. The best (not average) efficiencies obtained in this work are used for Seq. A (17.9%) and B (15.5%) to calculate $ Wp$^{-1}$. Since Seq. C is not experimentally demonstrated here, we have considered it with our highest recorded efficiency to date, achieved for a HTL-free and vacuum-free device (10%), as the most optimistic scenario. The figures clearly show the cost benefits of the carbon-electrode-based devices for both area and power-related cost metrics. The cost for Seq. B is likely to be lower than 1 USD W$^{-1}$, and Seq. C could be lower than 0.5 USD W$^{-1}$. These represent a significant reduction to the cost estimate from previous works of around 1.5 USD W$^{-1}$[47]. This results from a similar or lower cost in $ m$^{-2}$, and a higher recorded efficiency. However, the technology is still not able to compete with mass-produced silicon solar cells, for which module spot prices have been lower than 0.30 USD W$^{-1}$[48]. Despite this, opportunities may exist in niche markets that value the lightweight and flexible nature of these modules, as discussed in our previous work[47]. The next step for the technology would be exploring high-value PV markets at the predicted manufacturing

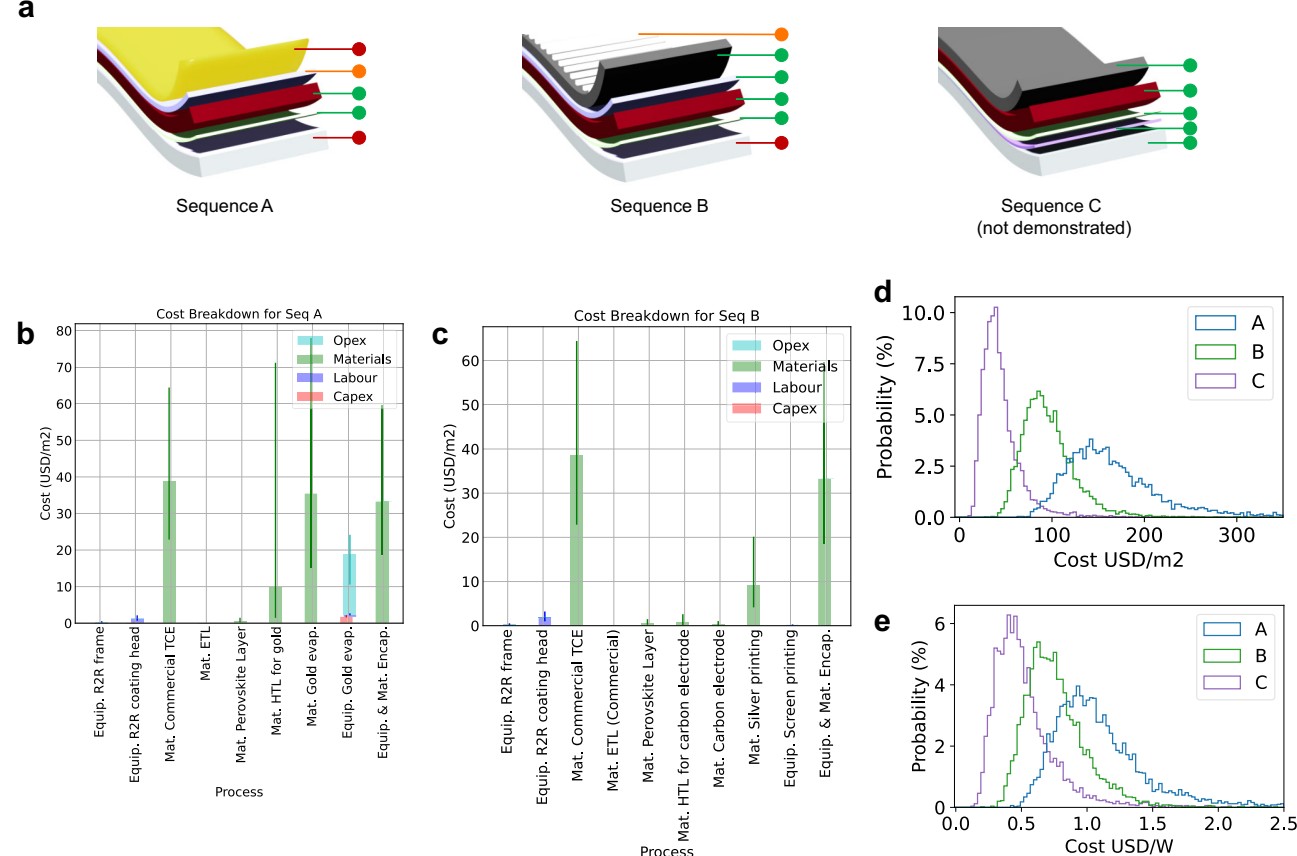

**Fig. 6 | Technoeconomic analysis of R2R-produced perovskite PV modules.**
**a** Device configurations used for the analysis. Sequence (seq.) A: high-cost and high-performance option, seq. B: fully R2R-fabricated device, and seq. C: ultra-low-cost printing option to be pursued. Cost breakdown for (**b**) seq. A and (**c**) seq. B. Projected production costs of the modules (**d**) per module area and (**e**) per peak watt for all three sequences based on 17.9%, 15.5%, and 10% PCE, respectively.

costs while addressing the remaining high-cost components to sustainably advance the technology towards commercialisation. Supplementary Fig. 12, with about 5 USD m⁻² module cost (excluding encapsulation), shows the potential for the further cost reduction by eliminating the remaining high-cost components.

In conclusion, we have successfully addressed the key challenges for low-cost roll-to-roll production of large-area perovskite solar modules and demonstrated the world-first fully roll-to-roll-fabricated perovskite solar modules (including back electrodes) on a commercial substrate. A perovskite-friendly carbon ink was developed to replace vacuum-processed metal electrodes, the highest-cost component in perovskite solar cells. This enabled the high-throughput, vacuum-free fabrication of perovskite solar cells using only roll-to-roll processes. Automated roll-to-roll fabrication and testing systems were developed to take full advantage of high-throughput fabrication, allowing thousands of research cells to be fabricated and tested in a single day to rapidly improve roll-to-roll experimentation. Further optimisation of the process and device configuration enabled fully roll-to-roll fabricated perovskite solar cells with up to 15.5% PCE, which represents the record efficiency for fully roll-to-roll fabricated perovskite solar cells to date. All developments in this work were performed with due consideration to upscaling, leading to the first demonstration of fully roll-to-roll printed perovskite solar modules with up to 11% PCE based on the active area of the module ~50 cm². Finally, the cost model developed in this work predicts the projected manufacturing cost of modules likely to be ~0.7 USD $W_p^{-1}$ with the potential for substantial further reduction via replacing remaining high-cost components with low-cost alternatives. This work demonstrates significant progress of the perovskite solar technology towards low-cost at-scale commercial manufacturing.

## Methods

### Materials
Materials for the preparation of flexible solar cells, lead(II) iodide (99.9985%) and tin(IV) oxide (15 wt% in $H_2O$ colloidal dispersion) were purchased from Alfa Aesar. MAI (99.99%), FAI (99.99%) and n-hexyl trimethyl ammonium bromide (HTAB) were purchased from Greatcell Solar Materials. Commercially available TCE substrates with a sheet resistance of 8 $\Omega$ sq⁻¹ (OPV8) were sourced from MekoPrint. Hole-transport materials poly[(2,5-bis(2-hexyldecyloxy)phenylene)-alt-(5,6-difluoro-4,7-di(thiophen-2-yl)benzo[c]-[1,2,5]thiadiazole)] (PPDT2FBT), poly(3-hexylthiophene) (P3HT, Lisicon SP001) and 2,2',7,7'-tetrakis(N,N-di-p-methoxyphenylamino)−9,9'-spirobifluorene (Spiro-MeOTAD) were purchased from 1-Materials, Merck and Luminescence Technology Corp. (Lumtec), respectively. FK209 was purchased from Lumtec. Silver paste (PV416) was purchased from DuPont. All other chemicals, including 4-tert-butylpyridine (t-BP), bis(trifluoromethane)sulfonimide lithium salt (LiNTf₂), dichlorobenzene (99%), anhydrous N,N-dimethylformamide (99.8%), and anhydrous 2-propanol (99.5%), and acetonitrile were sourced from Sigma-Aldrich and used as received.

### Ink Preparation for R2R coatings
To prepare stock solutions for reverse gravure (RG) coating, a mixture of 10 mL of a 15 wt.% solution of tin(IV) oxide and 5 mL of deionised water was used without any additives. The stock solution could be used for several RG coating trials over a few weeks. For SD coating, stock solutions were prepared by mixing 5 ml of the 15 wt% solution, 10 ml of deionised water and 15 μl of acetic acid. For the perovskite layer, a fresh PbI₂:FAI solution was prepared by dissolving 1.1 mmol (507 mg) PbI₂ and 0.45 mol% (0.5 mmol, 0.85 mg) FAI per 1 ml of

anhydrous N,N-dimethylformamide in a nitrogen-filled glove box, and stirred at 70 °C for approximately 1 h. The solution was cooled to ambient temperature before transferring to a SD head. The MAI solution for the second step of the deposition was made by stirring 40 mg of MAI per 1 ml of anhydrous 2-propanol for 10 min at ambient temperature. 1.0 mM of HTAB solution was prepared in a mixed solvent (chlorobenzene:isopropanol = 9:1 v/v%). The PPDT2FBT and P3HT HTM solutions were prepared by dissolving 10 mg of PPDT2FBT per 1 ml of dichlorobenzene and 5 mg of P3HT in 1 mL of dichlorobenzene, respectively. Polymers without any dopants were dissolved by stirring at 70 °C for more than 1 h. The Spiro-OMeTAD solution was prepared by mixing $6.0 \times 10^{-5}$ mol Spiro-OMeTAD (73 mg), $2.0 \times 10^{-4}$ mol t-BP (28.8 μL), $2.0 \times 10^{-4}$ mol LiNTf$_2$ (17 μL of 520 mg mL$^{-1}$ solution in CH$_3$CN), and $1.6 \times 10^{-6}$ mol FK209 (8 μL of 300 mg mL$^{-1}$ solution in CH$_3$CN) in 1 mL chlorobenzene.

## Carbon ink preparation

For the formulation of the SD ink, ethyl cellulose (EC, Sigma-Aldrich, 200646, viscosity 4 cP, 5% in toluene/ethanol) was used as the binder. A 1:1 mixture of carbon black (Vulcan XC72, Cabot) and graphene nanoplatelet powder (CamGraph G3, Cambridge Nanosystems) was used as the conductive carbon pigment. Propylene glycol methyl ether acetate (PGMEA) was used as the solvent. Due to the low viscosity of the desired SD ink, the ink preparation method was separated into two stages. In the initial stage, a high-viscosity ink was produced, which can be processed with a three-roll mill, breaking down pigment agglomerates into primary particles. For a 500 g ink batch, 40 g of EC was mixed in 330 g of PGMEA, stirred until the binder dissolved completely (Supplementary Fig. 9a). Later, 100 g of the solution was reserved for use in the second stage. The remaining 270 g of solution was mixed with 130 g of conductive carbon pigment to form a slurry (Supplementary Fig. 9b). This mixture was then processed through a three-roll mill, resulting in a uniform high viscosity ink (Supplementary Fig. 9c). In the second stage, the high viscosity ink was diluted to produce an ink that can be used for SD coating. The remaining 100 g of EC/PGMEA was hand-mixed with the high-viscosity ink and stirred with a magnetic stir bar until a uniform ink was obtained.

## Small-area cell fabrication

Functional layers were coated under ambient conditions on a benchtop R2R coater (Mini-Labo™, Yasui-Seiki) installed in a fume cabinet. A thin layer of SnO$_2$ ETL layer was coated on the flexible TCE substrate as received using the 10 Wt.% stock solution using reverse-gravure (RG) coating method at 4 rpm RG roll (200 R roll) speed, 0.25 m min$^{-1}$ line speed and a coating width of 13 mm. The wet film was found to dry immediately and was subsequently moved to a >135 °C curved hot plate for about 30 s, with hot air blowing at 120 °C for about 30 s.

The PET/TCE/SnO$_2$ film then underwent R2R IR treatment (2–3 W cm$^{-2}$) for about 5 min using an industrial R2R screen printer (Orthotec SRN3030). The film was then installed back onto the Mini-Labo coater for coating of the perovskite layer. The PbI$_2$:FAI solution was then SD coated (20 μL min$^{-1}$ flow rate, 0.3 m min$^{-1}$ web speed, 13 mm coating width) onto the SnO$_2$ film and the continuously moving wet film was then subjected to a flow of nitrogen using a 10 cm-wide air blade installed at the edge of supporting roller about 10 cm behind the coating head and 1–2 cm above the substrate. The N$_2$ flow rate was adjusted to about 50–100 L min$^{-1}$ to form a dried intermediate PbI$_2$:FAI layer. The MAI solution was then SD coated on the dried intermediate layer with the 60 μL min$^{-1}$ solution. Solvent evaporation was promoted by gentle air blowing using a small fan placed approximately 20 cm behind of the MAI coating head. The film was then passed over a hot plate at 135 °C for about 10 s. The PET/TCE/SnO$_2$/Perovskite film was then rewound and the HTAB and the P3HT solutions were deposited sequentially via SD coating at

0.3 m min$^{-1}$ line speed, first coating the HTAB layer (15 μL min$^{-1}$ flow rate, 7 mm coating width) followed by an annealing step on a curved hotplate at 100 °C for 30 s and then coating the P3HT layer (10 μL min$^{-1}$ flow rate, 6 mm coating width) by placing the SD head immediately above the second curved hotplate at 45 ± 5 °C. Finally, the carbon electrode was SD coated onto the P3HT layer using the PGMEA-based carbon ink by placing the SD head immediately above the curved hotplate at 70 °C (120 μL min$^{-1}$ flow rate and 5 mm coating width) to remove the solvents on the wet film completely, before an additional annealing step on the second curved hotplate at 130 °C. All R2R-processed devices were tested both with and without the additional screen-printed silver grid. The silver grid was screen printed using a semi-auto screen printer (Keywell KY-600FH) with 180 mesh screen onto the top carbon electrode and was annealed at 130 °C for 30 s on a hot plate.

For automatic fabrications, a thin SnO$_2$ ETL layer was prepared by SD coating of the 5 wt% solution (65 μL min$^{-1}$, 13 mm coating width) with an acetic acid additive at 0.2 m min$^{-1}$. (See Supplementary Note 1 for further details) The film was dried with hot air (135 °C for 5 min) and hot plate (135 °C for 1 min) using the commercial coater and then the film was IR treated as described above. 100 m length rolls are typically prepared and a batch of film could be used for multiple experiments over weeks. For other layers, the custom-built R2R machine shown in Fig. 3a was used. The same SD heads used in the commercial R2R coater were used in the custom-built machine. Therefore, all coating parameters were interchangeable between the two machines. However, a smaller air blade (13 mm width) with a lower flow rate (~20 L m$^{-1}$) was used for the nitrogen blowing. Besides the blowing condition, various coating parameters for PbI$_2$ and MAI solutions described in the main text were trialled with the automatic fabrication setup. For the fabrication of HTAB, P3HT, PPDT2FBT and carbon layers, the coating conditions described above were used in the custom-built machine. Once coating is complete, the roll at the winder was moved to the unwinder so that no rewinding was necessary. The two-step deposition for the perovskite layer was performed in a single pass and the HTAB/P3HT were also deposited in a single pass. So, the fabrication required total four coating runs (ETL, perovskite, HTL and carbon) and was typically completed within a day. The same-design SD heads could be used for all the layers. They were typically cleaned after full disassembly by wiping the remaining inks, followed by ultra-sonication in the solvent used in each ink for 5 min and drying by nitrogen blowing. The sonication and drying steps were repeated two more times, and the cleaning typically took 20 min. The tubing was used only once and disposed, except for the SD set for carbon layer (including tube, syringe and the ink), which was kept without disassembly in a sealed bag and used for multiple batches over months.

## Serially connected module fabrication

Functional layers were coated under ambient conditions on a benchtop R2R coater (Mini-Labo™, Yasui-Seiki) installed in a fume cabinet. All R2R-processed PeSC modules comprising five series-connected strip cells were fabricated on a stripe-patterned (13 mm stripes with 2 mm gap between stripes) commercial TCE. The module has the same configuration as the small cells, i.e., PET/TCE/SnO$_2$ FA$_{0.45}$MA$_{0.55}$PbI$_3$/HTAB/P3HT/Carbon/Ag. The SnO$_2$ ETL layer was deposited by RG coating in the same way as the small cells (200 R roll, 4 rpm, and 0.25 m min$^{-1}$ line speed) but with a roller with 5 stripes. The coating width was the same width as the TCE pattern so that the stripes were made with 0.5–1 mm of offset with respect to the patterned TCEs to achieve exposed TCE for series connections.

The remaining layers up to the carbon electrode were coated using the coating methods (SD coating head having five channels to deposit five 13 mm-wide wet coating stripes), line speeds, and the

annealing conditions illustrated in the previous section. Flow rates used for the coating of 5 stripes using the PbI$_2$:FAI, MAI, HTAB, P3HT and carbon inks were 100, 300, 140, 92, and 600 μL min$^{-1}$, respectively. The P3HT layer was deposited on a custom-built curved hot plate fitted with a heating tape (shown in Supplementary Fig. 4b) at 45 ± 5 °C. A screen-printed grid using a commercially available Ag ink was used to enhance the charge collection of the module and to interconnect the 5 cells in series. Screen printing was performed on an Orthotec-2 R2R screen printing system. The screen-printed Ag grid was dried using IR irradiance (1.5–2 W cm$^{-2}$) and hot air (90 °C). The active area of each strip cell was typically ~10 cm$^2$ (width: ~1.1 cm determined by manually controlled offset and length: 9.0 cm) resulting in an active module area of ~50 cm$^2$. The fabrication of R2R modules typically took two days and required two researchers.

## Characterisation methods

Manual J-V measurements were undertaken using a solar simulator (Newport Oriel) in air without encapsulation. The solar simulator was calibrated to 1-sun (1000 W m$^{-2}$) AM 1.5 G illumination using a certified Si reference cell with KG-1 filter (Enlitech, certified by Enlitech in accordance with IEC 60904-1:2006, spectral mismatch factor with carbon-based cells: 0.92) and a source metre (Keithley 2400). Devices were typically kept in air or a dry box for long-term storage before the measurement. A shadow mask was used to define a cell active area of 0.08 cm$^2$ for small cells. Modules were tested without a shadow mask. J-V measurements were carried out in the forward (increasing forward bias) and reverse (decreasing forward bias) scan directions over the voltage range from −0.2 V to 1.2 V with 20 mV step for cells (~250 mV s$^{-1}$) and −0.2 V to 5 V with 200 mV step (~1 V s$^{-1}$) for modules, respectively. For automatic J-V testing, a class AAA solar simulator (Enlitech SS-F5-3A) was used after calibration using the same reference cell. A secondary reference cell provided by Enlitech was also used to regularly check the light intensity. For the automatic testing, a circular aperture (1.8 mm diameter, Thorlabs) was used to define the beam size of 0.025 cm$^2$ which was illuminated on about 0.2 cm$^2$ cells defined by the coating width of carbon electrodes and the TCE pattern. The small aperture was deliberately chosen to tolerate the positioning error of the roll-to-roll tester. The PeSCs fabricated in the custom-built R2R coater were collected as a roll. The roll was then mounted on the R2R tester by positioning it on the winder of the tester and then rewound to get the starting point of the coating experiment. It is critical to set the position of the first cell under the aperture. A 2 cm gap in the TCE pattern was used to recalibrate the first cell position of every 20 cm block of 40 stripes by checking conductivity through the spring pins. The automatic testing was typically performed in air, but the data in Fig. 4c–g was obtained in a nitrogen-filled box. IPCE measurements were carried out using a commercial IPCE setup (Peccel S20). XRD patterns were obtained using a Rigaku SmartLab, equipped with a rotating anode CuKα source (45 kV, 200 mA), and Hypix 3000 detector. The SEM images of the films were taken with a Zeiss Merlin field emission SEM. A Hewlett-Packard 8453 diode-array spectrophotometer was used for optical density measurements. Time-resolved photoluminescence measurements were performed using a time-correlated single-photon counting (TCSPC) luminescence spectrometer (Edinburgh Instruments Ltd., FLSP920) comprising a pulsed diode laser excitation source (466 nm, 100 kHz, ~100 ps FWHM, ~0.2 nJ cm$^{-2}$ pulse$^{-1}$) and a Hamamatsu R928P photomultiplier tube detector, giving an overall instrument response time of ~0.7 ns (FWHM). Photoelectron Spectroscopy in Air (PESA) measurements were performed using a Riken Kekei AC2 spectrometer.

## Reporting summary

Further information on research design is available in the Nature Portfolio Reporting Summary linked to this article.

## Data availability

The experimental data that support the findings of this study are available in Figshare with the identifier: https://doi.org/10.6084/m9.figshare.24502210.

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

## Acknowledgements

This work was funded predominantly by the Australian Renewable Energy Agency (ARENA) (grant no. 2017/RND012). This work was also supported by the Australian Centre for Advanced Photovoltaics (ACAP) programme funded by the Australian Government through ARENA, the Australian Research Council Center of Excellence in Exciton Science (grant no. CE170100026), Australia-India Strategic Research Fund (AISRF) (grant no. AISRF75426), L.W.T.N. acknowledges a Nanyang Technological University (NTU) College of Engineering International Postdoctoral Fellowship. The authors acknowledge the assistance received from Dr Aaron Seeber with the analysis and interpretation of XRD results and CSIRO Manufacturing's Materials Characterisation and Modelling (MCM) team for SEM.

## Author contributions

H.C.W., N.M., and J.-E. K. contributed equally. H.C.W., J.-E.K., L.Sutherland, D.A., F.G. and D.V. fabricated cells and modules. N.M., L.W.T.N. and T.H. developed printable carbon pastes. D.V. developed automatic fabrication and testing system. A.S., L.Shi, A.W.Y.H.-B and A.S.R.C. performed characterisation of films and devices. N.C., M.D., R.E., R.C. and D.V. developed cost models and performed cost calculations. A.S.R.C., M.G., J.J.J., T. H. and D. V. coordinated and supervised this work.

## Competing interests

The authors declare no competing interests.
