## [Peer Review File · Nature Communications]

The first demonstration of entirely roll-to-roll fabricated perovskite solar cell modules under ambient room conditionsREVIEWER COMMENTS

Reviewer #1 (Remarks to the Author):

This a well thought out and comprehensive study representing a significant milestone in the manufacturing technologies associated with perovskite solar cells. It represents a very important step forward and includes a number of solutions to highly complex engineering challenges. I enjoyed the paper very much; it covers a wide range of technical challenges and introduces a number of new fabrication concepts. My particular highlights in the paper include

- the custom tester for measuring the modules in-line
- the edge blowing for removing chattering effects
- the cost analysis component
- the large parameter space measured

I propose some comments for the authors to consider but do not believe that they in-any way should prohibit publication.

- XRD Figure 2 - In the subset figure showcasing the PbI₂ peak, there is another smaller peak on the left of it, is that unconverted MAI? If so it would be useful to have it discussed in the text. Also, the subfigure does not have the labels.
- In the explanation of MAI optimisation by in-situ measurement, the language could be a little clearer. It's not very clear what parameters are being changed.
- A bit more explanation on how the substrate heating was done for P3HT deposition can be provided. How uniform the heating was? and how critical is uniform heating when making modules?
- Is it okay to use silver from a stability perspective since no stability data was provided? Does the carbon provide sufficient protection to migration.
- Regarding the tin oxide layer, the authors used acetic acid (as dopant), but its purpose was not clearly stated. KWas used to improve wetting or for any other specific reason?
- How did the authors managed to coat their solutions, particularly those containing highly toxic solvents like chlorobenzene, dichlorobenzene, and DMF, in an open area. Were there any concerns about exposure limits, and if so, how were they addressed?

- What was the g-FF of the modules (active v dead area), I think this is a critical addition as its likely to be low in comparison to typical silicon modules - this should be stated clearly and discussed and furthermore what values were inputted into the cost model.

I recommend acceptance of the article and offer congratulations to all the co-authors for a comprehensive and impactful paper.

Trystan Watson, Swansea University

Reviewer #2 (Remarks to the Author):

The work represents state of the art and a leading edge in the perovskite field for R2R printed flexible modules in air. The demonstration in and of itself is a noteworthy accomplishment and the fact that the PCEs are above 10% should be commended. I believe this manuscript is a good fit for Nature Communications, however, I do suggest the following be addressed before it is accepted:

Line 89 Fig 1a doesn't really illustrate a miniature "R2R factory" so much as it depicts components of developing a process, just a wording issue to fix. The wording is better in the figure capture, so just the line 89 text needs revisiting.

Fig 1c has a typo in the Iodide stoichiometry for the active layer.

Line 130 Edge blowing is a strange way to call an air knife blowing non-orthogonally (face blowing) to the substrate. Many people are doing this but don't call it edge blowing, just an angled gas quenching with an air blade/knife. Most people think about the edge in R2R processing as the edge of the substrate (either side/edge-on or the area of the web/substrate on the actual edges), so it's confusing since "edge" is used here since the step happens at the "edge" of a roller. Technically speaking, "edge blowing" here is still face blowing, just at an angle other than 90 degrees. I suggest making the language here more intuitive to the broader community, it will also avoid non-traditional language being brought into the field.

Line 196 – what is the L dimension of the coating from which the 80 cells were sourced? How much winding and unwinding did these cells experience and what is the bend radius?

Fig3a&b – the coating system and the testing system appear to be independent - is this true and if so please discuss the logistical challenges/influences of transferring the roll from one system over to the other? If not, please clarify this in the manuscript?

Why PPDT2FBT? Seems a random choice of the many semiconducting polymers (used in OPVs) that have been integrated with perovskites – the 2017 reference (40) does not go into any insightful details about the nature of interactions at the PPDT2FBT-PVSK interface, and none are offered here, unlike the 2019 ref (42) Nature paper using P3HT, where the HTAB side chain double layer alignment the 2D phase promotes templating of the polymer backbone using via hexyl side chain interactions with the hexyl chain on HTAB. Compared to P3HT, PPDT2FBT has much longer, and branched, alkoxy side chains on the phenyl unit. Why was it even used instead of P3HT-HTAB to begin with, and why not some other more strategically designed/chosen polymer based on rational/intentional molecular interactions at the HTL-PVSK interface?

Line 269 or 280 – Lane coating is commonly used in printing solar cells (OPVs, QDs, PVSKs), but produces notoriously low geometric FF's. The GFF of the printed modules is missing here, and should be explicitly stated/discussed.

Line 281 – the voltage scaling losses are also a major problem here. For the best cell, the Voc is 1.02 V, but for the best module, the Voc/cell = 4.59/5 = 0.918 V (~10% losses). Can you comment on the suspected reason for these losses. (i.e. – interconnect losses vs cell/printing damage losses).

What TCE is being used?

Line 282-283 What is the sheet resistance of the TCE, and what is the sheet resistance of the carbon electrode? It would be useful to justify the area-derived losses in FF being ascribed in part to the TCE (as opposed to and/or the carbon electrode).

Were you able to use different screen-printed mesh sizes to evaluate module performance impact (i.e. lower vs higher grid density), and how do the modules perform in the absence of the grid (i.e. – the Ag only printed in the interconnect, no extra current collection grid lines)? This savings in Ag printing content (if possible) could become important for the techno-economic analysis section, and since it's a small change from scenario B, can it be demonstrated?

I like the emphasis on how many cells can be fabricated and analysed in a day, but it would be more genuine to also include (in the SI) the entire time it takes for things like SD head setup, ink loading,

priming, web changing, clean-up (major one), etc... that are also involved in actually doing a research scale R2R PV module fab run.

It's not clear from the SI, the cartoon graphics, and the photographs, whether the modules have a silver mesh pattern, or a grid lines pattern. For example, Fig 1 seems to suggest the modules have a grid pattern, but the experimental section and Fig S5 are showing a mesh pattern. Please comment and correct if there is inconsistency.

Line 529 typo in MCM team name

Reviewer #3 (Remarks to the Author):

Translating the perovskite solar cell technology from lab-scale deposition techniques to the upscalable approaches is the next big step to be faced from the community. This challenge is tackled in the present work. I find the work interesting, well conducted and up to date. I propose the publication after some minor revisions are addressed.

1) The authors should discuss the advancement of their work with respect to the work of February 2023 published in advanced materials.

2) The authors states: "This strategy retards crystallisation and the precursor thin-film behaves like an amorphous material with much better film-forming properties than crystalline analogues." There are other approaches that can be used to retard the crystallisation of perovskite materials (such as the inclusion of functional additives (Nature Energy 7, 828 (2022), Nano Energy 54, 400 (2018), ACS Appl. Mater. Interfaces 2016, 8, 43, 29419–29426, Adv. Energy Sustainability Res. 4, 2200140 (2023), Adv. Energy Mater. 2023, 2203898 Nature 612, pages 266–271 (2022) etc.), some of which can also get rid of the solvent dripping/bathing step or of the double step. The authors should briefly discuss the use of additives as a possibility to retard the crystallisation of perovskite materials and perhaps also discuss some advantages of the proposed PFSD techniques with respect to the others.

3) The authors state "Supplementary Fig. 2 show signs of unreacted MAI on the face-blown sample". In order to be sure of this sentence a combined compositional/morphological analysis should be carried out. I would simply say that the sample obtained with edge blowing is more homogenous with compact grains.

4) Figure 4h. It can be observed a bump in the orange curve. Why is it so? Is it the electrode the problem here? can the authors show the forward and reverse curves for this device?

REVIEWER COMMENTS

Reviewer #1 (Remarks to the Author):

This a well thought out and comprehensive study representing a significant milestone in the manufacturing technologies associated with perovskite solar cells. It represents a very important step forward and includes a number of solutions to highly complex engineering challenges. I enjoyed the paper very much; it covers a wide range of technical challenges and introduces a number of new fabrication concepts. My particular highlights in the paper include

- the custom tester for measuring the modules in-line
- the edge blowing for removing chattering effects
- the cost analysis component
- the large parameter space measured

I propose some comments for the authors to consider but do not believe that they in-any way should prohibit publication.

We sincerely appreciate the reviewer's constructive feedback and thoughtful comments on our manuscript. In response to these comments, we have provided a comprehensive and detailed response below.

- XRD Figure 2 - In the subset figure showcasing the PbI_2 peak, there is another smaller peak on the left of it, is that unconverted MAI? If so it would be useful to have it discussed in the text. Also, the subfigure does not have the labels.

We appreciate the reviewer's comment and suggestion. The peak at 10.4° corresponds to a hydrated perovskite complex, as previously identified (Reference: <https://doi.org/10.1039/C7TA02264F>). Such a complex is expected in air-exposed samples. The small peak at 11.8° is consistent with $\delta\text{-FAPbI}_3$, as noted in (Reference: <https://doi.org/10.1039/C7CE00402H>). Figure 2c has been updated to include this information.

- In the explanation of MAI optimisation by in-situ measurement, the language could be a little clearer. It's not very clear what parameters are being changed.

We appreciate the reviewer's comment. The discussion has been revised as below for added clarity.

Line 192: "Three PbI_2 conditions were selected to fabricate perovskite layers of about 600 nm to 1000 nm thickness. This range is somewhat thicker than typical vacuum-deposited electrode devices due to the absence of a mirror effect from the carbon-based back electrode. Also, a perfectly matched stoichiometry is not necessarily the best formulation in the $\text{FA}_{0.45}\text{MA}_{0.55}\text{PbI}_3$ system, as it can benefit from either a slight excess of lead⁴⁴⁻⁴⁶ or a cation-excessive composition.⁴⁷ Therefore, the ability of SD coating to give quantitative control over the amount of material deposited allowed for the amount of MAI present in the perovskite layer to be varied from slightly cation deficient (lead-excessive composition) through to stoichiometric and slightly

excessive compositions for each PbI_2 condition. The MAI-deposition flow rate was varied between 30-100 $\mu\text{L min}^{-1}$ in 10 $\mu\text{L min}^{-1}$ intervals.”

- A bit more explanation on how the substrate heating was done for P3HT deposition can be provided. How uniform the heating was? and how critical is uniform heating when making modules?

We sincerely thank the reviewer for raising these questions. In our slot die coating process, we utilised curved hot plates for the coating stages. The cylindrical coating stages beneath the slot die heads shown in Figure S1a and S1b are equipped with heating elements and temperature sensors, enabling precise temperature control.

For the modules, we employed a curved metal plate fitted with a heating tape and a thermocouple. We have included a photograph of this setup in Figure S4b for reference.

It is worth noting that we indeed found heating to be essential (noticeable improvement from 35°C) to achieve uniform P3HT layer deposition on top of HTAB, as shown in the new Figure S4c below. However, we would like to emphasise that the precise heating temperature appears to be less critical. The heater incorporates a single sensor covering a relatively large area, but during our experiments we encountered no issues related to temperature uniformity or temperature fluctuations within the range of 40-50°C.

In addition to the new figures (Figure S4b and S4c), we also added further details on P3HT coating in the Methods section.

- Is it okay to use silver from a stability perspective since no stability data was provided? Does the carbon provide sufficient protection to migration.

We acknowledge the reviewer's concern regarding the suitability of silver as a material for use in perovskite PV. Regrettably, we currently lack definitive information on the long-term durability of printed silver grids for potential commercial applications. To address this important question, we intend to conduct long-term studies. However, our future efforts will focus on the development of high-conductivity perovskite-friendly carbon inks, allowing us to create silver-free modules. Therefore, we added the following comments in the manuscript.

Line 328: “However, the printed silver used in this work may not be suitable for long-term operation in commercial applications due to corrosion. The next challenge is to develop a perovskite-friendly conductive carbon ink that is at least as conductive as TCEs to produce efficient silver-free PeSC modules.”

- Regarding the tin oxide layer, the authors used acetic acid (as dopant), but its purpose was not clearly stated. Was used to improve wetting or for any other specific reason?

We employed acetic acid to enhance the wetting properties without altering the characteristics of the dried tin oxide films. It's noteworthy that we encountered a significant challenge related to de-wetting of the tin oxide solution during the R2R fabrication process. To provide comprehensive details on this issue, we dedicated a separate section, Supplementary Note 1, in the SI due to space limitations.

In order to ensure visibility of this critical information, we have added the following statement to the Methods section for the tin oxide: "(See Supplementary Note 1 for further details) "

We trust that this clarifies our approach and addresses any oversight regarding the supplementary note.

- How did the authors managed to coat their solutions, particularly those containing highly toxic solvents like chlorobenzene, dichlorobenzene, and DMF, in an open area. Were there any concerns about exposure limits, and if so, how were they addressed?

We extend our gratitude to the reviewer for bringing attention to this vital matter. The safety protocols associated with our research hold utmost significance, and we are fully dedicated to the ongoing development of a safe operating protocol, particularly as we proceed with upscaling our work.

Fortunately, the commercial benchtop coater employed in this study possessed a compact design, enabling us to conduct all coating experiments within the confines of our fume cabinets. We have incorporated this pertinent information into the manuscript.

Methods section: "Functional layers were coated under ambient conditions on a benchtop R2R coater (Mini-Labo™, Yasui-Seiki) installed in a fume cabinet."

- What was the g-FF of the modules (active v dead area), I think this is a critical addition as its likely to be low in comparison to typical silicon modules – this should be stated clearly and discussed and furthermore what values were inputted into the cost model.

We appreciate the reviewer's comment. The g-FF (defined as the active area over the total area between two contact electrodes) is approximately 75%, and we used 85% (expected upper limit for the stripe coating approach: <https://doi.org/10.1002/aenm.202100342>) for the cost model. For the cost calculation, we chose realistic parameters for commercial production rather than what we used in this work as discussed in the Supplementary Note 4.

Although we made our best effort to describe the scenarios, we recognize that we omitted such critical information. We have added a paragraph to address this topic in the manuscript and an additional sentence in SI, as outlined below.

Line 301: "The interconnection of cells was formed through gaps between the stripe patterns at the same time as the grid. The active area is 49.5 cm² (1.1 cm × 9 cm × 5 cells) and the geometric fill factor (GFF), as defined as the cell area over total area (cell area + gap area)⁵² of the modules, is 75%. The GFF is somewhat lower than that of laser scribed modules with a demonstrated GFF of up to 99%⁵³ due to the inherent limitation of the stripe-pattern approach. However, it's worth noting that the laser scribing process may not be suitable for the high-throughput, cost-effective manufacturing of PeSCs. Therefore, our next challenge lies in developing modules with higher GFFs and larger areas while continuing to enhance cell efficiencies through scalable processes."

Supplementary note 4: "We also used an 85% geometric fill factor (GFF) expected upper limit³² and a realistic number for commercial products fabricated by industrial printers, rather than the 75% GFF demonstrated in this work with the lab machine with a basic web guide system."

I recommend acceptance of the article and offer congratulations to all the co-authors for a comprehensive and impactful paper.

Trystan Watson, Swansea University

We are genuinely thankful to Prof. Watson for the recommendation.

Reviewer #2 (Remarks to the Author):

The work represents state of the art and a leading edge in the perovskite field for R2R printed flexible modules in air. The demonstration in and of itself is a noteworthy accomplishment and the fact that the PCEs are above 10% should be commended. I believe this manuscript is a good fit for Nature Communications, however, I do suggest the following be addressed before it is accepted:

We sincerely appreciate the reviewer's constructive feedback and thoughtful comments on our manuscript. In response to these comments, we have provided a comprehensive and detailed response below.

Line 89 Fig 1a doesn't really illustrate a miniature "R2R factory" so much as it depicts components of developing a process, just a wording issue to fix. The wording is better in the figure capture, so just the line 89 text needs revisiting.

We thank to the reviewer for the good suggestion. The sentence has been revised as below.

Line 89: "This was achieved by developing: (i) a robust and scalable deposition technique, (ii) perovskite-friendly carbon inks to replace vacuum-based electrodes, and (iii) a R2R-based high-throughput experimental platform as illustrated in Fig. 1a."

Fig 1c has a typo in the Iodide stoichiometry for the active layer.

We thank the reviewer for pointing out the error. It has been fixed.

Line 130 Edge blowing is a strange way to call an air knife blowing non-orthogonally (face blowing) to the substrate. Many people are doing this but don't call it edge blowing, just an angled gas quenching with an air blade/knife. Most people think about the edge in R2R processing as the edge of the substrate (either side/edge-on or the area of the web/substrate on the actual edges), so it's confusing since "edge" is used here since the step happens at the "edge" of a roller. Technically speaking, "edge blowing" here is still face blowing, just at an angle other than 90 degrees. I suggest making the language here more intuitive to the broader community, it will also avoid non-traditional language being brought into the field.

We thank the reviewer for the suggestion. We revised the manuscript with "right-angle blowing" and "shallow-angle blowing" instead of "face blowing" and "edge blowing", respectively. We hope the new terms are acceptable.

Line 196 – what is the L dimension of the coating from which the 80 cells were sourced? How much winding and unwinding did these cells experience and what is the bend radius?

We appreciate the reviewer for highlighting this point. The substrate has approximately 4 mm wide electrode patterns, as shown in Figure 3c, with a repeating pattern every 5 mm over a length of 20 cm, resulting in 40 cells per block. Each block is separated by a 2 cm gap, making each condition cover a length of 44 cm. This number has been added to the Figure 3 caption.

CSIRO

Australia's National Science Agency

During fabrication, the cells go through four winding stages, which include ETL, active layer (two coatings at a time), HTL (two coatings at a time), and the carbon layer. The smallest bend radius encountered during this process was 12.5 mm, facilitated by the rubber roller that drives the web. The information has been added to the Methods section.

Methods section: "Once coating is complete, the roll at the winder was moved to the unwinder so that no rewinding was necessary. The two-step deposition for perovskite layer was performed in a single pass and the HTAB/P3HT were also deposited in a single pass. So, the fabrication required total four coating runs (ETL, perovskite, HTL and carbon) and typically completed within a day. The same-design SD heads could be used for all the layers. They were typically cleaned after full disassembly by wiping the remaining inks, followed by ultrasonication in the solvent used in each ink for 5 min and drying by nitrogen blowing. The sonication and drying steps were repeated two more times, and the cleaning typically took 20 min. The tubing was used only once and disposed, except for the SD set for carbon layer (including tube, syringe and the ink), which was kept without disassembly in a sealed bag and used for multiple batches over months."

Fig3a&b – the coating system and the testing system appear to be independent - is this true and if so please discuss the logistical challenges/influences of transferring the roll from one system over to the other? If not, please clarify this in the manuscript?

The coating system and testing system are two independent systems. After the fabrication process is complete, we get the solar cells in a roll form on the winder (the roll on the right side in Figure 3a). Subsequently, we install the roll onto the winder of the tester and manually pull the film, attaching it to the unwinder with an empty core. This practice aligns with the standard procedure for all roll-to-roll systems and presents no challenges. The only critical thing is ensuring the first cell is properly positioned at the testing stage when we commence the testing process. We have added further details of the testing in the Methods section as below.

Methods section: "The PeSCs fabricated in the custom-built R2R coater were collected as a roll. The roll was then mounted on the R2R tester by positioning it on the winder of the tester and then rewound to get the starting point of the coating experiment. It is critical to set the position of the first cell under the aperture. A 2 cm gap in the TCE pattern was used to recalibrate the first cell position of every 20 cm block of 40 stripes by checking conductivity through the spring pins."

Why PPDT2FBT? Seems a random choice of the many semiconducting polymers (used in OPVs) that have been integrated with perovskites – the 2017 reference (40) does not go into any insightful details about the nature of interactions at the PPDT2FBT-PVSK interface, and none are offered here, unlike the 2019 ref (42) Nature paper using P3HT, where the HTAB side chain double layer alignment the 2D phase promotes templating of the polymer backbone using via hexyl side chain interactions with the hexyl chain on HTAB. Compared to P3HT, PPDT2FBT has much longer, and branched, alkoxy side chains on the phenyl unit. Why was it even used instead of P3HT-HTAB to begin with, and why not some other more strategically designed/chosen polymer based on rational/intentional molecular interactions at the HTL-PVSK interface?

We acknowledge the reviewer's inquiry regarding our choice of materials. It is essential to understand that only a small fraction of reported materials and processes are suitable for roll-to-roll (R2R) fabrication, primarily due to variations in solvent drying kinetics and the intricate requirements of the R2R process. Therefore, it is necessary to screen those materials/processes and optimise the inks for R2R fabrication. So, we must settle on a material that exhibits reasonable performance, reliability, and processability to facilitate the development of other components of the device.

Having the improved performance with HTAB-P3HT data in this work, it seems that the HTL system should have been the first choice. However, even the research group that reported HTAB-P3HT (Jung et al. ref.50) did not use this HTL in their R2R printed perovskite paper (Kim et al. ref. 25). This might be attributed to challenges in delicately controlling HTAB deposition or the lack of substantial improvements in their trials. We could achieve improved performance only after significant efforts (the optimisation was carried out with a different recipe so only the final optimum condition is reported here).

CSIRO

Australia's National Science Agency

At the time of screening/developing perovskite-friendly carbon inks, the polymer was internally the best HTL for R2R fabrication. So, we used the materials as a standard HTL during development. The PPDT2FBT polymer was initially developed for organic PV, showed exceptional thickness tolerance (i.e. good charge transporting performance) despite the complex molecular structure (<https://doi.org/10.1039/C4EE01529K>, <https://doi.org/10.1002/adfm.201505556>), and has been regarded as a promising material for R2R printing. We also confirmed R2R processability of the material by fabricating organic PV by R2R slot die coating (Song et al. ref 46). The background was briefly mentioned in the Supplementary Note 3. We expanded the background in the note as below and also added a sentence "(further discussion on the material choice can be seen in Supplementary Note 3)" in the manuscript.

Supplementary Note 3: "Spiro-OMeTAD was found to be not suitable for printed electrodes due to its low thermal durability, as thermal annealing over at least 120°C is essential to dry carbon and silver inks. At that stage, poly(3-hexylthiophene) (P3HT) was not an option as we previously found that P3HT alone does not perform as well as some other available HTLs⁹. We also tried commercially available HTL solutions developed for PeSCs, such as Clevios HTL Solar 3 and Solar 4. These HTL solutions are commonly used poly(3,4-ethylenedioxythiophene) (PEDOT) dissolved in a perovskite compatible organic solvent (toluene for Solar 3 and anisole for Solar 4). Solar 3 was first reported by Hou et al.²⁵ and used in a recent report on the fully R2R fabricated PeSCs¹⁴. We also tested both materials and found they work better than P3HT, however, we also found poly[(2,5-bis(2-hexyldecyloxy)phenylene)-alt-(5,6-difluoro-4,7-di(thiophen-2-yl)benzo[c][1,2,5]-thiadiazole)] (PPDT2FBT) slightly better than the commercial HTL solutions.

PPDT2FBT was first developed for organic solar cells²⁶ and was found to be extremely thickness tolerant²⁷. Despite its complex molecular structure, intramolecular hydrogen bonds lock the planarity of aromatic units and enhance molecular packing, which ensures good charge transportation performance. Therefore, the material has been regarded a printable material (sold as "Solar Polymer for R2R Printing") and we confirmed suitability for R2R printing by demonstrating record-breaking efficiency²⁸ (at the time of the publication) for R2R fabricated organic solar cells. The polymer was also used as an HTL and has shown high performance and durability in PeSCs²⁹. Therefore, we chose the polymer as a standard material while developing other parts of PeSCs."

Line 269 or 280 – Lane coating is commonly used in printing solar cells (OPVs, QDs, PVSKs), but produces notoriously low geometric FF's. The GFF of the printed modules is missing here, and should be explicitly stated/discussed.

We thank the reviewer for the comment and added following sentences in the manuscript.

Line 301: "The interconnection of cells was formed through gaps between the stripe patterns at the same time as the grid. The active area is 49.5 cm² (1.1 cm × 9 cm × 5 cells) and the geometric fill factor (GFF), as defined as the cell area over total area (cell area + gap area)⁵² of the modules, is 75%. The GFF is somewhat lower than that of laser scribed modules with a demonstrated GFF of up to 99%⁵³ due to the inherent limitation of the stripe-pattern approach. However, it's worth noting that the laser scribing process may not be suitable for the high-throughput, cost-effective manufacturing of PeSCs. Therefore, our next challenge lies in developing modules with higher GFFs and larger areas while continuing to enhance cell efficiencies through scalable processes."

Line 281 – the voltage scaling losses are also a major problem here. For the best cell, the Voc is 1.02 V, but for the best module, the Voc/cell = 4.59/5 = 0.918 V (~10% losses). Can you comment on the suspected reason for these losses. (i.e. – interconnect losses vs cell/printing damage losses).

The performance of the modules, which have an active area approximately 2000 times larger than small cells, is expected to align more closely with that of average devices rather than the champion cell. As depicted in

Figure 4f, the typical voltage hovers around 0.9 V. It is important to acknowledge that a certain degree of performance loss is inevitable during the upscaling process.

As discussed in the manuscript, we attribute the efficiency loss primarily to a reduction in the fill factor (FF), stemming from the elevated resistance of the transparent conducting electrode (TCE). While this resistance measures $8 \Omega \text{ sq}^{-1}$, it is insufficiently high for the 9.9 cm^2 cell, and it becomes a prominent contributing factor to the efficiency decline. Additionally, the screen printing of the silver grid is also responsible for the current loss, as indicated in Figure S11.

What TCE is being used?

Line 282-283 What is the sheet resistance of the TCE, and what is the sheet resistance of the carbon electrode? It would be useful to justify the area-derived losses in FF being ascribed in part to the TCE (as opposed to and/or the carbon electrode).

We employed a commercially available substrate known as OPV8 substrate (with a sheet resistance of $8 \Omega \text{ sq}^{-1}$) sourced from MekoPrint, as detailed in the Methods section. The sheet resistance of the carbon electrodes is approximately $800 \Omega \text{ sq}^{-1}$ (this information has been included in both the Methods section and Fig. S11). However, it's important to note that the silver grids exhibit higher conductivity than the TCE and therefore the conductivity of the back electrode would not be the limiting factor.

Were you able to use different screen-printed mesh sizes to evaluate module performance impact (i.e. lower vs higher grid density), and how do the modules perform in the absence of the grid (i.e. – the Ag only printed in the interconnect, no extra current collection grid lines)? This savings in Ag printing content (if possible) could become important for the techno-economic analysis section, and since it's a small change from scenario B, can it be demonstrated?

The aim of the experiment for Fig. S11 was to find out the answer to the reviewer's question (but cells were used instead of modules). The cell without a grid design exhibited notably lower performance compared to those with the printed silver grid. However, as the loading of the printed silver was increased, the performance of the devices showed a decline. This performance loss is attributed to the damage of underlying layers caused by the solvent present in the silver ink. Consequently, it is essential to incorporate additional conductors while minimizing the use of silver ink.

To achieve this, we opted for a 180 mesh, which is the finest mesh that allowed us to reliably print the silver paste without encountering clogging issues. Additionally, we used a 0.2 mm line width, the narrowest width that we could consistently print. Since we have already minimized the silver ink usage in this work, further reductions in production costs for Scenario B would prove challenging. Our next step is to explore a more cost-effective alternative, as outlined in Scenario C.

Recognizing the absence of discussion regarding grid design, we have included the following discussion in the manuscript.

"It was crucial to incorporate additional conductive elements alongside the carbon layer, which had a sheet resistance of approximately $800 \Omega \text{ sq}^{-1}$. Supplementary Fig. 11 illustrates that cells without a grid design exhibited significantly poorer performance compared to those with grids. Nonetheless, excessively covering the screen-printed silver led to a decline in performance, most likely due to solvent damage to the underlying layers. Consequently, we designed the silver pattern to achieve minimal coverage while maintaining adequate conductivity, at least surpassing that of the front electrode, for efficient charge collection. We determined that a 0.2 mm line with a 180 mesh screen provided the finest pattern that we could consistently print onto the carbon surface, and this parameter was adopted for the module."

I like the emphasis on how many cells can be fabricated and analysed in a day, but it would be more genuine to also include (in the SI) the entire time it takes for things like SD head setup, ink loading, priming, web changing, clean-up (major one), etc... that are also involved in actually doing a research scale R2R PV module fab run.

For the small cells, which feature compact dimensions of 25 mm × 40 mm as shown in Figure S3, the setup process, including ink loading, priming, and web setup, takes approximately 5-10 minutes, and cleaning is conveniently conducted while coating another layer. Consequently, the cleaning of the slot die head typically requires about 20 minutes, but it does not extend the overall fabrication time as the SD head unit (including tubing and syringe) for the last layer (carbon) was typically kept together with the ink in a sealed bag and used without cleaning for the next batch.

It's worth noting that we have demonstrated (to be published) the fabrication of more than 10,000 organic photovoltaic cells (including ink preparation, setup, and clean-up) within a 24-hour timeframe by one person, all tested within the following 24 hours. This underscores the genuinely high-throughput nature of our system.

In this work, we did not find a meaningful study for such many parameter combinations. So, we only mentioned the proven capability of the platform. But we can confirm that a batch of experiment (including setup and cleaning) can be completed in a day.

Further description on the high-throughput cell fabrication and module fabrication has been added to the Methods section.

"The same-design SD heads could be used for all the layers. They were typically cleaned after fully dissembled by wiping remaining inks followed by ultrasonication in the solvent used in each ink for 5 min and drying by nitrogen blowing. The sonication and drying were repeated two more times, and the cleaning typically took 20 min. The tubing was used only once and disposed. But the SD set for carbon layer including a tube, a syringe and the ink was kept without dissembling in a sealed bag and used for multiple batches over months. The fabrication R2R PeSCs on top of SnO₂ coated TCE could be completed in a day."

"The fabrication of R2R modules typically took two days and required two researchers."

It's not clear from the SI, the cartoon graphics, and the photographs, whether the modules have a silver mesh pattern, or a grid lines pattern. For example, Fig 1 seems to suggest the modules have a grid pattern, but the experimental section and Fig S5 are showing a mesh pattern. Please comment and correct if there is inconsistency.

We apologize for any confusion caused. We would like to confirm that the modules indeed feature a grid pattern, and the schematic drawing is accurate. We acknowledge that the photograph presented in Figure S5 may not provide a clear depiction due to digital artifacts. To address this, we have updated the figure to include a zoom-in section as illustrated below.

Line 529 typo in MCM team name

We thank for the reviewer for pointing that out, but we confirm "Dr Aaron Seeber" is correct name.

Reviewer #3 (Remarks to the Author):

Translating the perovskite solar cell technology from lab-scale deposition techniques to the upscalable approaches is the next big step to be faced from the community. This challenge is tackled in the present work. I find the work interesting, well conducted and up to date. I propose the publication after some minor revisions are addressed.

We are grateful to the reviewer for the positive feedback and for raising important points. We have provided a detailed response to all the comments below.

1) The authors should discuss the advancement of their work with respect to the work of February 2023 published in advanced materials.

The work by Watson et al. (Reference 29) marked a significant milestone in the field by demonstrating the first fully roll-to-roll (R2R) fabricated perovskite solar cells (PeSCs) through the development of R2R-processable and perovskite-friendly carbon inks. While this work was pivotal in terms of potentially reducing the manufacturing cost of PeSCs, it achieved an efficiency of only 10.8% at a cell area of 0.09 cm² (defined by an optical aperture).

Our work significantly increased the efficiency of fully R2R fabricated PeSCs, largely attributed to the introduction of the HTAB-P3HT hole transport layer (HTL) system. Additionally, we are proud to report the successful fabrication of fully R2R fabricated large-area PeSC modules for the first time. This achievement was made possible by developing an extremely reliable deposition process under ambient conditions. The following discussion has been incorporated into the manuscript to provide further context.

Line 80: "Due to these technical challenges, the first example of a small-area PeSC (0.09 cm² active area) having all layers deposited on a flexible plastic substrate using R2R processes was only very recently reported (in February 2023)²⁹ with individual cells displaying PCEs of up to 10.8%. While the first report marked a significant milestone in the field, the efficiency was still far from that of typical research cells and only small cells were demonstrated.

Here we report the fabrication of entirely R2R-printed individual PeSCs with a record-high 15.5% PCE. We also report the first demonstration of PeSC modules produced using only industry-relevant R2R fabrication techniques, and under ambient room conditions."

2) The authors states: "This strategy retards crystallisation and the precursor thin-film behaves like an amorphous material with much better film-forming properties than crystalline analogues." There are other approaches that can be used to retard the crystallisation of perovskite materials (such as the inclusion of functional additives (Nature Energy 7, 828 (2022), Nano Energy 54, 400 (2018), ACS Appl. Mater. Interfaces 2016, 8, 43, 29419–29426, Adv. Energy Sustainability Res. 4, 2200140 (2023), Adv. Energy Mater. 2023, 2203898 Nature 612, pages 266–271 (2022) etc.), some of which can also get rid of the solvent dripping/bathing step or of the double step. The authors should briefly discuss the use of additives as a possibility to retard the crystallisation of perovskite materials and perhaps also discuss some advantages of the proposed PFSD techniques with respect to the others.

We thank the reviewer for the comment. Indeed, the additive approaches have been helpful to get R2R fabricated PeSCs without using double-step fabrication or solvent dripping. We have explored various additive approaches (surfactants, salts, 2D cations, and polymers) together with the hot deposition technique (ref. 31-36). However, we still found the PFSD technique most reliable and suitable to produce large-area modules. Therefore, we decided to further develop the double-step deposition technique although it is not ideal to have the additional step. The discussion has been added to the manuscript as below.

"The introduction of the printing-friendly sequential deposition (PFSD) technique by select co-authors of this work in 2017³⁰ enabled the demonstration of the first PeSCs comprising R2R-deposited electron-transport

CSIRO

Australia's National Science Agency

layer (ETL), light-absorbing layer, and hole-transport layer (HTL), with up to 11% PCE achieved for a small-area device. Since then, we also developed more facile single-step deposition techniques via the introduction of various additives, such as polymers, ammonium salt, and 2D organic cations, together with heating and nitrogen blowing^{31–36}, and investigated R2R techniques reported by others^{37–41}. Although it was possible to produce the perovskite layer in a single-step deposition, we found no approach that significantly outperforms PFSD for R2R-based upscaling.”

3) The authors state “Supplementary Fig. 2 show signs of unreacted MAI on the face-blown sample”. In order to be sure of this sentence a combined compositional/morphological analysis should be carried out. I would simply say that the sample obtained with edge blowing is more homogenous with compact grains.

We thank the reviewer for the suggestion and revised the manuscript as suggested.

4) Figure 4h. It can be observed a bump in the orange curve. Why is it so? Is it the electrode the problem here? can the authors show the forward and reverse curves for this device?

It may appear that there is a bump in the JV curve, possibly attributed to the blue curve in the figure. However, we confirm that there is no bump in the curve other than measurement noise. To enhance clarity, we have introduced a grid in the figure, as shown below.

REVIEWERS' COMMENTS

Reviewer #1 (Remarks to the Author):

Very happy that the authors have addressed my comments in full. Really great work!

Trystan Watson

Reviewer #2 (Remarks to the Author):

All my concerns have been addressed and I appreciate the quality of the revision.

PS - The typo I referenced in the Acknowledgements section was not referring to Dr. Seeber's name, but rather: "CSIRO ***Manufacturing's*** Materials Characterisation and Modelling (MCM) team for SEM"

Reviewer #3 (Remarks to the Author):

The authors have addressed the reviewers' comments. The paper can be accepted for publication in Nature Communications.